# Seeking and receiving help for mental health services among pregnant women in Ghana

Samuel Adjorlolo  *

Department of Mental Health Nursing, School of Nursing and Midwifery, College of Health Sciences, University of Ghana, Legon Accra, Ghana

* sadjorlolo@ug.edu.gh

## Abstract

### Objective

The heightened vulnerability of women to mental health issues during the period of pregnancy implies that seeking and receiving support for mental health services is a crucial factor in improving the emotional and mental well-being of pregnant women. The current study investigates the prevalence and correlates of seeking and receiving help for mental health services initiated by pregnant women and health professionals during pregnancy.

### Design

Using a cross-sectional design and self-report questionnaires, data were collected from 702 pregnant women in the first, second and third trimesters from four health facilities in the Greater Accra region of Ghana. Data were analyzed using descriptive and inferential statistics.

### Results

It was observed that 18.9% of pregnant women self-initiated help-seeking for mental health services whereas 64.8% reported that health professionals asked about their mental well-being, of which 67.7% were offered mental health support by health professionals. Diagnosis of medical conditions in pregnancy (i.e., hypertension and diabetes), partner abuse, low social support, sleep difficulty and suicidal ideation significantly predicted the initiation of help-seeking for mental health services by pregnant women. Fear of vaginal delivery and COVID-19 concerns predicted the provision of mental health support to pregnant women by health professionals.

### Conclusion

The low prevalence of individual-initiated help-seeking implies that health professionals have a high responsibility of supporting pregnant women achieve their mental health needs.

**Data Availability Statement:** The data underlying the results presented in the study are available from DOI: 10.17632/bmgbndy38g.2.

**Funding:** The project received financial support awarded to Samuel Adjorlolo from the Canadian

Queen Elizabeth II Diamond Jubilee
Advanced Scholars Program (QES-AS). The QES-
AS is made possible with financial support from
IDRC and SSHRC. Website of funding support:
https://www.univcan.ca/programs-and-
scholarships/queenelizabeth-scholars/qes-
advanced-scholars/ The funders had no role in
study design, data collection and analysis, decision
to publish, or preparation of the manuscript.

**Competing interests:** The authors have declared
that no competing interests exist.

## Introduction

There is ample literature regarding the burden of mental health problems during the periods of pregnancy (i.e., prenatal period) and after delivery (postnatal period); a burden that is highly elevated in low- and middle-income countries mainly due to the preponderance of risk factors such as poverty, food insecurity, pollution, migration, conflict, climate change, and gender-based violence [1, 2]. Depression, for instance, is estimated to affect about 11.3% of pregnant women and 18.9% of mothers after birth in Africa, whereas the prevalence of pre- and postnatal anxiety has been estimated at 14.8% and 14%, respectively [3]. Perinatal mental illness has been associated with adverse pregnancy outcomes such as preterm birth, intrauterine growth restriction [4] and childhood behavioral and emotional problems [5, 6]. These adverse pregnancy outcomes have been implicated in maternal and child mortality in Ghana [7].

While the provision of mental health support is central to improving the mental well-being of pregnant women [8, 9], this is also contingent on help-seeking for mental health services. Rickwood, Deane [10] conceptualized help-seeking as a social transaction between intrapersonal and interpersonal processes. The intrapersonal process is concerned with an individual's level of awareness and demonstration of the need to seek help. This process is premised on the assumption that an individual has the capacity and competency to differentiate among different mental states and seek help when they perceive or experience mental health issues. The awareness of this intrapersonal need is expected to lead to a *self-initiated help-seeking* for mental health services. Under the self-initiated help-seeking framework, pregnant women are expected to activate the mental healthcare pathway by initiating contact with and seeking help from (mental) health professionals to manage their emotional problems. Self-initiated mental health help-seeking largely attests to the willingness of pregnant women to share their emotional problems with others as well as the desire to seek and receive help.

However, research has shown that pregnant women are less likely to seek support for mental health on their own for reasons such as fear of being shamed, embarrassed, socially isolated and stigmatized [11–14]. Other barriers include lack of information about services and access to care, lack of knowledge regarding maternal mental health and preference for informal sources of support [14]. The stigmatization of mental health and psychiatric facilities in Ghana, for instance, is among the incentives pushing individuals to seek mental health support from informal sources, including religious leaders and traditional healers [15]. Therefore, health professionals have important roles in ameliorating the emotional and mental well-being of pregnant women [8, 9]. Indeed, evidence-based clinical and practice guidelines directs health professionals to prioritize and promote the mental health of pregnant women. The United Kingdom's National Institute for Health Care Excellence (NICE) guidelines categorically places the responsibility of maternal mental health in the hands of health professionals. These professionals are expected to initiate conversations around pregnancy and mental health, to enquire about the emotional well-being of pregnant women and offer mental health support, including referral services [16]. The American college of Obstetricians and Gynecologists (ACOG) patient safety bundle on maternal mental health has similarly tasked the health delivery system and professionals to proactively take steps to improve the wellbeing of pregnant women. Some recommended measures include establishing a response protocol and identifying screening tools, gathering data via administering valid screening tools and activating the necessary response system for mental health support. Such interactions can lead to a reduction in the stigma associated with perinatal mental health, while serving as an avenue to address other common barriers such as lack of knowledge, attitudes and beliefs that affect maternal mental health [14]. Research has shown that pregnant women are more likely to seek help for their mental health issues when health professionals take steps to foster trust, enquire about perinatal mental health issues, and offer the needed support [12, 17].

Although perinatal mental health has gained traction in the literature [14, 18], no study has systematically explored pregnant women self-initiating help-seeking for mental health services as well as health professionals' involvement in maternal mental health simultaneously. Previous studies have centered on help-seeking among pregnant women diagnosed with or experiencing mental health challenges [13, 19, 20]. There is a general lack of evidence on help-seeking among pregnant women in general. Extending the discussions and empirical literature to all pregnant women, rather than focusing on those diagnosed with mental disorders, is consistent with the call for a population approach to promoting mental health [21, 22] and the recommendation by the UK's NICE and ACOG patient safety bundle on maternal mental health. This approach also has the opportunity to increase not only mental health literacy in the pregnancy period but also empower pregnant women to proactively safeguard their mental well-being through initiatives such as reducing maternal stressors and seeking help for emotional discomfort. Together, these can have a considerable impact on the health and well-being of their children and the next generations, given the relationship between maternal mental well-being and infant, child and adolescent development and psychopathology [5, 6, 21].

In addition to the limited studies focusing on self-initiated mental health help-seeking among pregnant women [13, 19, 20], the involvement of health professionals in promoting maternal mental health by, first and foremost, enquiring about pregnant women's emotional well-being and, secondly, providing mental health support, including referral services, needs to be examined. Although recent studies have explored challenges affecting the delivering of perinatal mental health services by health professionals [8, 9], it is not clear the extent to which these professionals are supporting pregnant women in meeting their emotional well-being needs. Moreover, the range of factors that is influencing self-initiated mental health help-seeking among pregnant women and health professionals' involvement in promoting maternal mental health is unclear. If known, they could form the basis for intervention programming.

Lastly, there is limited empirical evidence on seeking and receiving help for mental health services in pregnancy among women in low- and middle income countries, including Ghana, despite the reported high burden of mental illness [23]. Management of mental health issues in Ghana mostly involve a mixture of psychiatric and faith-based and traditional healing services [15]. Evidence suggests that faith-based and traditional healing services are mostly the first point of contact because of a confluence of factors [15]. These include ascribing spiritual meanings to mental health issues and the uneven distributionof psychiatric services across the 16 regions in Ghana [24, 25]. Efforts to decentralize and integrate the provision of mental health services into primary healthcare is underway in regional and district hospitals across the country. Owing to their supposed complementarity, discussions have been advanced for a working collaboration between biomedical, faith and traditional healing pathways for the management of mental disorders [15]. Thus, pregnant women, like other patients, could invoke psychiatric-hospital treatment pathways or the faith-based and traditional healing services. The Ghanaian health professionals, like their counterparts in high-income countries, have a responsibility of ensuring and responding to the emotional and well-being issues of their clients/patients. In addition to providing basic mental health services such as screening and counseling, nurses and midwives have the responsibility to refer clients/patients suspected of mental disorders to mental health professionals for support and care. Seeking help for and receiving mental health support from health professionals would help to improve the mental well-being of pregnant women. However, as noted previously, there is limited data on pregnant women seeking help on their own as well as receiving mental health support from health professionals in Ghana and other LMICs.

Research addressing these topical issues is needed to provide the evidence base to develop a holistic strategy to ensure that pregnant women with mental health challenges have the

appropriate services and support systems required. Consequently, the overarching goal of the study is to investigate mental health help-seeking among pregnant women by addressing the aforementioned research gaps. The study objectives were: (1) to investigate the number of pregnant women who initiated mental health help-seeking on their own, who were asked about their mental well-being and/or were offered mental health support by health professionals and (2) to determine the influence of factors culled from medical history, intimate partner violence, COVID-19 worries, social support, fear of delivery, sleep difficulty and suicidal ideation domains on self-initiated help-seeking and health professional involvement in mental health support.

## Methods

### Ethical approval

The study received ethics clearance from the Noguchi Memorial Institute for Medical Research, University of Ghana (NMIMR-IRB CPN 057/19-20).

### Participants

The data were collected as part of a study investigating maternal mental health in Ghana. Data were gathered from 702 pregnant women recruited from the antenatal clinics of four health facilities, namely University of Ghana hospital ($n = 175$, 24.9%), Alpha hospital ($n = 239$, 34%), Sanford Clinic ($n = 87$, 12.4%) and St-Gregory Catholic hospital ($n = 201$, 28.6%) in the Greater Accra and Central regions of Ghana. These facilities, located across different geographical areas, are patronized by pregnant women of different demographic backgrounds.

### Study design and data collection

A cross-sectional design was used. Participants were recruited from the health facilities based on their availability and willingness to participate in the study. The inclusion criteria for participating in the study were age ≥18 years, and completion of senior high school diploma or equivalent (i.e., 12 years of formal education). This requirement assured the participants were proficient in English. It was most practical to use English questionnaires as there are more than 10 languages spoken in the Greater Accra and Central regions of Ghana. The exclusion criteria were diagnosis of mental disorder during pregnancy and history of mental health disorder. Data were collected by research assistants (RAs) who received training in salient issues relating to research with pregnant women, including privacy, confidentiality, non-judgmental attitude, recruitment, and questionnaire administration. At each facility, nurses or midwives were identified as facility-based focal persons. These individuals supported the recruitment process by introducing the research assistants to the participants attending the antenatal clinics. Thereafter, the RAs approached and discussed the study with individual pregnant woman. Participants were recruited on based on convenience. The participants, who often congregate at the antenatal clinics of the facilities, were informed about the purpose and duration of the study, their responsibilities for participating in the study, ethical issues such as confidentiality, consent, anonymity and benefits of participation. Questions raised by the participants were responded to by the research team to allay fears, anxieties and encourage participation in the study.

Prior to completing the questionnaire, the participants read and signed the consent form, together with the RAs. The "broad consent" gave the research team the permission to obtain data on pregnancy outcomes and other pertinent obstetric information after delivery from the participants' folders, where necessary. The folder/hospital identity numbers of the participants

were recorded to facilitate subsequent matching of information. The questionnaires were completed individually and independently. The RAs were present to provide the needed support to the participants. Once completed, the questionnaires were handed over to the RAs. Data collection lasted for approximately eight weeks, from September to October 2020. The COVID-19 precautionary measures such as wearing of nose masks and use of alcohol-based hand sanitizers were strictly adhered to.

### Data collection measures

**Outcome variables.** The outcome variables for the study were self-initiated help-seeking and health professional involvement in maternal mental. Self-initiated help-seeking was measured with a single item: *I have sought help for emotional or mental health issues while pregnant in the last 3 months*. Based on the recommendations from the NICE (2014) and ACOG (2016) guidelines, the participants responded to two items to estimate the extent to which health professionals were involved in promoting their mental well-being: (1) *Nurses and midwives have asked me about my mental health or emotional well-being* and (2) *Nurses or midwives have offered me support such as counselling and referral for my mental health or emotional well-being needs*. The response options for self-initiated help-Seeking and health professional involvement were as follows: Never; Yes, once; Yes, twice and Yes, several times. These were subsequently recoded into two categories for each item: "Never" (coded 0) and "at least once" (coded 1).

### Predictor variables

A total of seven variables were considered as predictor variables in this study. These were described below.

**Obstetric history.** The participants were asked whether they have been diagnosed with medical conditions such as hypertension and diabetes in their previous pregnancy and current pregnancy. Responses were dichotomous: Yes or No.

**Fear of delivery:** This was measured with two items: 1) Fear of vaginal delivery and (2) fear of caesarean section. The participants responded by choosing either "Yes" or "No".

**Intimate partner violence:** This was operationalized using two items from the literature [26, 27]: (1) I have been belittled by my partner (i.e., abuse) and (2) I experienced no attention from my partner (i.e., neglect). Responses were dichotomous: Yes or No.

**COVID-19 concerns.** The literature has revealed heightened worries and fears about the coronavirus in the general population [28]. Among pregnant women, this fear could be extended to the impact of contracting the coronavirus on the pregnancy and baby. We therefore assessed COVID-19 concerns using two items that were scored on a four-point Likert response scale from Not all (0) to Very often (3). The first item relates to whether the participants were worried about contracting the virus and the second involved whether the participants were worried that their babies could develop some birth or developmental abnormalities should they contract the coronavirus. A total score was obtained by summing the responses, with higher scores indicating more COVID-19 concerns. A Cronbach's Alpha of 0.92 was obtained for the two-item "COVID-19 concern" scale.

**Sleep difficulty.** This was measured by asking about (1) difficulty to fall asleep while in bed and (2) difficulty to stay asleep through the night. The items were extracted from the existing literature [29] and were scored using a four-point Likert response format ranging from Not at all (0) to Very often (3). Responses to each item were added to create a total score, with higher scores indicating more sleep difficulty. The Cronbach's Alpha for the sleep difficulty scale was 0.86.

**Suicidal ideation.** Following a review of the literature [30], three items were extracted to index suicidal ideation as follows: Have you (1) ever thought that life wasn't worth living; (2) ever thought about killing yourself; and (3) ever attempted to kill yourself? The response options ranged from Never (0) to Yes, several times (3). Total scores obtained by summing the responses on the scale ranged from 0 to 9, with higher scores reflecting more suicidal ideation. The Cronbach's Alpha for the suicidal ideation scale was 0.71.

**Social support.** This was also measured with three items culled from the existing literature [26] as follows: (1) Someone is around when I am in need; (2) I have someone with whom I can share my joy and sorrow; and (3) I have received emotional support from my partner. The participants responded to the items using a four-point Likert scale, from Strongly disagree (1) to Strongly agree (4). The responses were added together with total score ranging from 3 to12. Higher scores indicated more social support. A Cronbach's Alpha of 0.87 was recorded for the Social Support Scale.

**Data analysis strategy.** Data were analyzed using SPSS Version 23 (IBM.corp), with a two-tailed statistical significance set at 0.05. The outcome variables were analyzed using descriptive statistics, namely frequency and percentages. Chi-square was used to investigate the bivariate relationship between the outcome variables and categorical predictor and demographic variables. The relationship between outcome and continuous predictor variables (e.g., age, social support, COVID-19 worries, sleep difficulty and suicidal ideation) were examined using point-biserial correlation. The point-biserial correlation was used to examine the relationship between a continuous predictor variable and a dichotomous outcome variables. The coefficient for point-biserial correlation ranges from -1 to 1 and is interpreted in the same way as Pearson or Spearman correlation coefficients. Lastly, binary logistic regression was used to examine the predictors of the outcome variables. In the first of the series of logistic regression analysis, the regression model contained only one predictor. This was followed by the model of individual predictors controlling for demographic variables. Lastly, all the predictor variables were entered into the model concurrently, controlling for demographic variables. Results were reported as crude odds ratios (COR) and adjusted odds ratios (AOR) with 95% confidence intervals (95% CI). Only the variables that demonstrated significant bivariate correlations with the outcome variables were included in the binomial logistic regression.

## Results

### Participants

The participants were recruited from the various pregnancy periods (1st trimester = 63, 9%; 2nd trimester = 315, 44.9% and 3rd trimester = 324, 46.2%). Out of the total participants recruited, 125 (18%) and 569 (82%) were single and married, respectively. More than half (*n* = 358, 53%) completed senior high school/equivalent (at least 12 years of education), 168 (24.9%) completed post-secondary school (additional 2 or 3 years of schooling from senior high school) whereas 22.1% (*n* = 149) completed university education. In terms of employment, 48.6% (*n* = 322) reportedly worked in government or private establishments (so-called formal employment), 29.6 (*n* = 196) in informal employment and 21.8% (*n* = 144) were unemployed. The average age of the participants was 30 years (*SD* = 5.54).

Intimate partner violence appeared highly prevalent, with 36.4% (*n* = 686) and 86% (*n* = 686) of the study participants reported partner abuse and neglect, respectively. A total of 34.2% (*n* = 634) and 66.8% (*n* = 633) participants were fearful of vaginal delivery and delivery by caesarean section, respectively. Data on obstetric history showed that 12.7% (*n* = 693) of the participants were diagnosed with medical conditions, namely hypertension and diabetes while pregnant in the past, whereas14.5% (*n* = 691) were diagnosed with these conditions in the present pregnancy.

## Prevalence of mental health help-seeking

It was observed that 18.9% of the participants initiated mental health help-seeking on their own, suggesting majority ($n$ = 557, 81.1%) did not seek mental health support in the last three months prior to the study (Table 1). In contrast, more than half ($n$ = 444, 64.8%) of the participants reported that nurses and midwives enquired about their mental and emotional wellbeing at least once. In terms of support, 47.1% ($n$ = 320) of the participants reported that they were offered mental health support by the nurses and midwives. A statistically significant correlation was observed between being asked about mental wellbeing and being offered mental health support ($\chi^2$ = 209.11, p < .001). About 67.7% of the participants who were asked about their mental wellbeing were offered some mental health support.

When analyzed across the various demographic factors, varying results were obtained (Table 2). For instance, self-initiated mental health help-seeking did not differ significantly across the trimesters of pregnancy. However, participants in the second trimester were more likely to have been asked about their mental well-being (z = 2.2, $p$ < 0.01) and/or received some mental health support from health professionals (z = 3.0, $p$ < 0.001). Health professionals were also more likely to ask about mental well-being or provide mental health support to participants who completed only senior high school.

## Intercorrelations among study variables

Table 3 contained a summary of the results of the intercorrelations among the study variables. As can be seen, the study variables were significantly correlated. The outcome variables correlated with several of the predictor variables. Self-initiated mental health help-seeking correlated significantly with past medical history ($\chi^2$ = 34.48, p < .01), present medical history ($\chi^2$ = 27.97, p < .01), partner neglect ($\chi^2$ = 11.24, p < .01), partner abuse ($\chi^2$ = 26.32, p < .01), social support ($r$ = -.36, p < .01), sleep difficulty ($r$ = .21, p < .01) and suicidal ideation ($r$ = .31, p < .01). Health professionals were more likely to ask about mental well-being and provide mental health support to participants diagnosed with medical conditions in pregnancy (e.g., hypertension), those who feared vaginal delivery, were victims of partner abuse, had difficulties sleeping and harbored suicidal ideations.

## Predictors of mental health help-seeking

Different sets of binary logistic regressions were performed for each of the outcome variables. The results of the model containing the individual predictors and model of individual predictors controlling for demographic factors are shown in Table 4. The majority of the variables individually and independently predicted the outcome variables. For example, participants who experienced partner abuse were 2.73 ($\chi^2$ = 25.19, $p$ < 0.001), 1.61 ($\chi^2$ = 7.61, $p$ < 0.01) and 1.56 ($\chi^2$ = 7.64, $p$ < .01) more likely to seek mental health support on their own, to have

**Table 1. Prevalence of self and health professional-initiated help-seeking for mental health among pregnant women (n = 702).**

| Variables/Response | Never | Yes | Once | Twice | Several times |
|---|---|---|---|---|---|
| | $n$ (%) | $n$ (%) | $n$ (%) | $n$ (%) | $n$ (%) |
| **Self-Initiated Help-Seeking** | | | | | |
| Mental health seeking in last 3 months (**$n$ = 687**) | 557(81.1) | 130(18.9) | 83(12.1) | 30(4.4) | 17(2.5) |
| **Health Professional-Initiated Help-Seeking** | | | | | |
| Asked about mental health issues (**$n$ = 685**) | 241(35.2) | 444(64.8) | 227(33.1) | 101(14.7) | 116(16.9) |
| Offered mental health support (**$n$ = 679**) | 359(52.9) | 320(47.1) | 160(23.6) | 78(11.5) | 82(12.1) |

**Table 2. Frequencies and Chi Square results for demographic correlates of the outcome variables.**

| Source | Self-Initiated Mental Health Help Seeking | | | Health Professional Initiated Mental Health Help Seeking | | | Health Professional Providing Mental Health Support | | |
|---|---|---|---|---|---|---|---|---|---|
| | No n (%) | Yes n (%) | Value | No n (%) | Yes n (%) | Value | No n (%) | Yes n (%) | Value |
| **Age+** | - | - | .01, ns | - | - | .03, ns | - | - | .14** |
| **Research Setting** | | | 96.07*** | | | 87.76*** | | | 208.11*** |
| Legon | 144 (21) | 28(4.1) | | 75(10.9) | 93(13.6) | | 122(18.0) | 45(6.6) | |
| Alpha | 210(37.7) | 23(17.7) | | 122(17.8) | 110(16.1) | | 174(25.6) | 55(8.1) | |
| Sanford | 37(5.4) | 49(7.1) | | 18(2.6) | 68(9.9) | | 38(5.6) | 46(6.8) | |
| St-Gregory | 166(24.2) | 30(4.4) | | 26(3.8) | 173(25.3) | | 25(3.7) | 174(25.6) | |
| Total | 557(81.1) | 130(18.9) | | 241(35.2) | 444(64.8) | | 359(52.9) | 320(47.1) | |
| **Trimester** | | | .39, ns | | | 23.96*** | | | 31.70*** |
| First | 52(7.6) | 11(1.6) | | 25(3.6) | 36(5.3) | | 36(5.3) | 25(3.7) | |
| Second | 245(35.7) | 61(8.9) | | 77(11.2) | 228(33.3) | | 124(18.3) | 179(26.4) | |
| Third | 260(35.8) | 58(8.4) | | 139(20.3) | 180(26.3) | | 199(29.3) | 116(17.1) | |
| Total | 557(81.1) | 130(18.9) | | 241(35.2) | 444(64.8) | | 359(52.9) | 320(47.1) | |
| **Education** | | | 4.40, ns | | | 46.65*** | | | 48.80*** |
| SHS/below | 276(41.6) | 72(10.9) | | 102(15.5) | 249(37.7) | | 154(23.5) | 194(29.7) | |
| Diploma or equivalence | 133(20.1) | 33(5) | | 43(6.5) | 121(18.3) | | 79(12.1) | 84(12.8) | |
| First degree & above | 129(19.5) | 19(2.9) | | 85(12.9) | 60(9.1) | | 112(17.1) | 31(4.7) | |
| Total | 538(81.3) | 124(18.7) | | 230(34.8) | 430(65.2) | | 345(52.8) | 309(47.2) | |
| **Marital Status** | | | 16.37*** | | | 3.02, ns | | | 1.84, ns |
| Single | 86(12.6) | 39(5.7) | | 35(5.2) | 89(13.1) | | 58(8.6) | 65(9.7) | |
| Married | 469(68.9) | 87(12.8) | | 202(29.8) | 352(51.9) | | 296(44) | 253(37.6) | |
| Total | 555(81.5) | 126(18.5) | | 237(35) | 441(65) | | 354(52.7) | 318(47.3) | |
| **Employment** | | | 2.91, ns | | | .44, ns | | | 8.22* |
| Formal | 251(38.4) | 69(10.6) | | 113(17.4) | 201(30.9) | | 179(27.8) | 134(20.8) | |
| Informal | 157(24) | 33(5.1) | | 65(10) | 129(19.8) | | 85(13.2) | 107(16.6) | |
| Unemployed | 121(18.5) | 22(3.4) | | 52(8) | 90(13.8) | | 76(11.8) | 63(9.8) | |
| Total | 529(81) | 124(19) | | 230(35.4) | 420(64.6) | | 340(52.8) | 304(47.2) | |

* p < .05

** p < .01

*** p < .01, ns = not significant.

been asked about their mental well-being and to have been offered mental health support by health professionals, respectively.

Adjusted odds ratios were obtained after controlling for demographic variables that evinced significant effect on the outcome variables (Table 2).

When all the predictors were entered into the model simultaneously while controlling for demographic variables, the results, summarized in Table 5, showed that the history of diagnosis of medical condition in a previous pregnancy (AOR = 3.80, $p < 0.01$), partner abuse (AOR = 1.88, $p < .05$), social support (AOR = 0.80, $p < 0.001$), sleep difficulty (AOR = 1.35, $p < .01$) and suicidal ideation (AOR = 1.22, $p < .01$) emerged as the significant predictors of self-initiated mental health help-seeking. None of the variables significantly predicted the likelihood that participants were asked about their mental well-being by health professionals. In contrast, fear of vaginal delivery (AOR = 1.91, $p < .05$) and COVID-19 concerns (AOR = 1.19, $p < .05$) were significantly associated with health professionals offering mental health support to the participants.

**Table 3. Chi Square and point-biserial correlation interrelations results for study variables.**

|  | 1 | 2 | 3 | 4 | 5 | 6 | 7 | 8 | 9 | 10 | 11 | 12 |
|---|---|---|---|---|---|---|---|---|---|---|---|---|
| **Self-Initiated Help-Seeking** | | | | | | | | | | | | |
| **1.** Last 3 month | 1 | | | | | | | | | | | |
| **Professional Initiated Help-Seeking** | | | | | | | | | | | | |
| **2.** Asked | 27.72** | 1 | | | | | | | | | | |
| **3.** Support offered | 18.31** | 209.11** | 1 | | | | | | | | | |
| **Medical History** | | | | | | | | | | | | |
| **4.** Past | 34.48** | 5.39* | 14.16** | 1 | | | | | | | | |
| **5.** Present | 27.97** | 10.07** | 13.26** | 320.79** | 1 | | | | | | | |
| **Fear of Delivery** | | | | | | | | | | | | |
| **6.** Vaginal delivery | 27.08** | 16.98** | 50.28** | 1.03 | .63 | 1 | | | | | | |
| **7.** Cesarean Section | 2.01 | 8.60** | 26.04** | 2.76 | .01 | 28.24** | 1 | | | | | |
| **Intimate Partner Violence** | | | | | | | | | | | | |
| **8.** Partner Neglect | 11.24** | .46 | .51 | 11.34** | 13.40** | 2.48 | .30 | 1 | | | | |
| **9.** Partner abuse | 26.32** | 7.67** | 7.67** | 11.30** | 15.13** | 3.90* | 17.39 | .40 | 1 | | | |
| **10.** Social Support+ | -.36** | -.08* | -.03 | .14** | .21** | .14** | .05 | .25** | -.31** | 1 | | |
| **11.** COVID-19+ concern | .09* | .26*** | .41** | -.21*** | -.13** | -.26** | -.33** | -.04 | .17** | .06 | 1 | |
| **12.** Sleep Difficulty+ | .21** | .22*** | .31** | -.10** | -.12** | -.30** | -.24** | -.11** | .14** | -.08* | .47** | 1 |
| **13.** Suicidal Ideation+ | .31** | .11** | .08* | -.09* | -.12** | -.17** | -.07 | -.21** | .16** | -.39** | .05 | .17** |

* $p < .05$

** $p < .01$

+ = point-biserial correlation coefficients.

Demographic variables that evinced significant effect on the outcome variables (Table 2) were controlled for.

**Table 4. Predictors of mental health help-seeking.**

|  | Self- Help-seeking | | Asked about Mental health | | Offered Mental Health Support | |
|---|---|---|---|---|---|---|
|  | OR (95% CI) | AOR (95% CI) | OR (95% CI) | AOR (95% CI) | OR (95% CI) | AOR (95% CI) |
| **Medical history** | | | | | | |
| Past | 3.90***(2.419–6.289) | 3.58***(2.071–6.184) | 1.83*(1.093–3.079) | 1.18(0.639–2.158) | 2.42***(1.511–3.870) | 1.21(0.608–2.412) |
| Present | 3.13**(1.970–4.979) | 2.26**(1.328–3.847) | 2.25**(1.349–3.74) | 1.67(0.922–3.021) | 2.24***(1.439–3.474) | 1.40(0.736–2.669) |
| **Fear of delivery** | | | | | | |
| Vaginal delivery | 2.91***(1.926–4.395) | 1.94**(1.186–3.185) | 2.17***(1.496–3.157) | 1.13(0.714–1.788) | 3.47***(2.444–4.938) | 2.13**(1.254–3.603) |
| C- section | 1.38(0.883–2.156) | 1.29(0.770–2.153) | 1.68**(1.185–2.371) | 1.01(0.672–1.504) | 2.45***(1.728–3.463) | 1.17(0.721–1.912) |
| **Partner violence** | | | | | | |
| Partner neglect | .44**(0.274–720) | .47*(0.266–838) | 1.17(0.746–1.829) | 1.38(0.807–2.351) | .85(0.553–1.319) | .87(0.461–1.655) |
| Partner abuse | 2.73***(1.845–4.045) | 2.12*(1.362–3.301) | 1.61**(1.148–2.260) | 1.12(0.753–1.667) | 1.56**(1.139–2.145) | 1.11(0.697–1.750) |
| **Social support** | .70***(0.648-.762) | .77***(0.695–845) | .93*(0.869-.996) | .95(0.867–1.034) | .98(0.920–1.043) | .97(0.869–1.071) |
| **COVID-19** | 1.10*(1.014–1.202) | 1.15*(1.017–1.035) | 1.30***(1.205–1.411) | 1.09(.987–1.211) | 1.51***(1.391–1.631) | 1.21**(1.082–1.359) |
| **Sleep difficulty** | 1.38***(1.227–1.548) | 1.37***(1.191–1.586) | 1.36***(1.221–1.515) | 1.15*(1.011–1.305) | 1.52**(1.364–1.686) | 1.18*(1.020–1.363) |
| **Suicidal ideation** | 1.61***(1.415–1.839) | 1.40***(1.208–1.619) | 1.21(1.056–1.378) | 1.16(0.997–1.349) | 1.13*(1.006–1.261) | 1.12(0.954–1.303) |

* $p < 0.05$

** $p < 0.01$

*** $p < 0.001$; C–section = Caesarean section

**Table 5. Predictors of mental health help-seeking, controlling for individual predictors and demographic variables.**

| | Help-seeking last 1 month | Asked about Mental health | Offered Mental Health Support |
|---|---|---|---|
| | AOR (95% CI) | AOR (95% CI) | AOR (95% CI) |
| **Medical history** | | | |
| Past | 3.80** (1.636–8.842) | .74(.320–1.706) | .82(.317–2.127) |
| Present | 0.75 (0.326–1.747) | 1.97(.858–4.509) | 1.24(.496–3.088) |
| **Fear of delivery** | | | |
| Vaginal delivery | 1.59(.903–2.792) | 1.07(.662–1.743) | 1.91*(1.097–3.312) |
| Caesarean section | .78(.422–1.433) | .82(.526–1.275) | 1.06(.630–1.768) |
| **Partner violence** | | | |
| Partner neglect | .74(.367–1.471) | 1.46(.775–2.757) | .85(.392–1.823) |
| Partner abuse | 1.88*(1.091–3.288) | 1.11(.703–1.750) | 1.11(.655–1.882) |
| **Social support** | .80***(.711–908) | .96(.863–1.063) | .95(.842–1.080) |
| **COVID-19 worries** | 1.07(.905–1.261) | 1.09(.971–1.234) | 1.19*(1.038–1.354) |
| **Sleep difficulty** | 1.35**(1.125–1.626) | 1.04(.896–1.201) | 1.05(.881–1.239) |
| **Suicidal ideation** | 1.22**(1.092–1.449) | 1.13(.950–1.339) | 1.09(.916–1.307) |

\* $p < .05$

\*\* $p < .01$

\*\*\* $p < .001$

# Discussion

Mental health challenges are common among pregnant and postpartum women across the globe. However, studies investigating mental health help-seeking behaviors for this vulnerable population are limited globally and virtually non-existent in Ghana and other low- and middle-income countries.

## Self-initiated mental health help-seeking in pregnancy

Since the period of pregnancy is associated with physiological changes and the likelihood of exposure to stressful factors such as financial difficulties and occupational hazards that have been implicated in perinatal mental health [27, 31, 32], changes in emotional and mental well-being are expected. It is postulated that pregnant women who initiate mental health help-seeking process by themselves are more likely to have the ability not only to detect changes in their emotional state and mental well-being but more importantly to act on these changes by seeking help and keeping to a treatment schedule. In this study, only a small number of pregnant women reportedly sought help on their own in the last three months prior to the study. This finding is consistent with an earlier finding [11]. In addition to the social (e.g., stigma and embarrassment), instrumental (e.g., financial restraints) and structural barriers (e.g., lack of information) affecting help-seeking for mental health services by pregnant women [14], the limited access to mental health services in Ghana [24] could contribute to the low help-seeking for mental health services among pregnant women. The provision of mental health services is concentrated at the three public psychiatric institutions in the country, two of which are located in the Greater Accra region of Ghana. Owing to stigma, most people are not willing to visit these facilities for support. Although there are ongoing efforts to decentralize and integrate the provision of mental health services into primary healthcare [33], the limited publicity of these services means that several people, including pregnant women, may not be aware of their availability. The traditional world view of Ghanaians regarding demons or supernatural

forces as causes of mental disorders places additional constraints on pregnant women's willingness to initiate mental health help-seeking from friends and family members and health professionals, despite the documented benefits of doing so [13].

The study showed that few pregnant women who reportedly sought help on their own for mental health problems were more likely to be women with obstetric history, were victims of intimate partner violence, had sleep difficulty, experienced suicidal ideation and reported low social support. This observation largely supports the existing literature on the relationship between exposure to risk factors and utilization of mental health services [27, 31, 32].

## Health professionals involvement in promoting mental health in pregnancy

The observation that pregnant women infrequently seek mental health support on their own volition as reported in this and previous studies [11] calls for more pragmatic efforts from health professionals. In this study, more than half of the pregnant women reported that nurses/midwives asked about their emotional or mental well-being. This observation is largely consistent with an earlier study in Ghana in which more than half of the nurses and midwives reported that they were involved in promoting maternal mental health such as asking pregnant women about their emotional well-being [9]. Demographically, pregnant women in the second trimester and those with a high school diploma or equivalent were more likely to be asked about their mental and emotional well-being. Anecdotally, most pregnant women in Ghana begin attending antenatal clinics in their second trimester. This raises the possibility that a comprehensive intake assessment undertaken for first time antenatal clinic attendees may comprise an assessment of their mental and emotional well-being. The documented relationship between low education and risk for poor mental health [34] could partly explain why pregnant women with secondary school education in this study reported that health professionals enquired about their emotional and mental well-being.

When demographic variables were controlled for, sleep difficulty emerged as the robust predictor of the pregnant women being asked about mental and emotional well-being as well as being offered mental health support such as counselling and referral services. Poor sleep not only negatively impacts mental health [35], but it is also a symptom of poor mental health (Anderson & Bradley, 2013). Therefore, sleep difficulty is likely to come to the attention of health professionals as a contributory factor to the mental health challenges reported by pregnant women. Pregnant women who feared vaginal delivery and expressed concerns about COVID-19 were reportedly offered mental health support. In a population-based cohort study, fear of vaginal delivery was strongly associated with a preference for elective caesarean section [36]. Estimated to range from 7%-27%, elective caesarean section is a major concern among researchers, policy-makers, stakeholders and health professionals owing to the associated surgical complications [37] and perhaps costs and consequences for future pregnancies. Efforts to persuade pregnant women capable of vaginal delivery to do so may involve offering them mental health support, including counselling and providing referral sources to allay fear and anxiety. A similar argument could be advanced for concerns regarding COVID-19. That is, health professionals may offer mental health support to reduce the fears and anxiety surrounding COVID-19 as a way of encouraging the continuous use of antenatal services and to promote delivery at health facilities in the COVID-19 pandemic.

Lastly, nurses and midwives enquired about the mental health or emotional wellbeing of more than half of the participants. In terms of support, about 67% of the participants received some mental health support. Perhaps, those who received support were those who indicated poor mental health or emotional distress to the nurses and midwives. There is also the possibility that some pregnant women may not be offered mental health support, although they may

indicate problems with their mental health or emotional well-being. While the reasons accounting for this were not explored in this study, a previous study identified barriers hampering the involvement of nurses and midwives in perinatal mental health in Ghana. These include lack of knowledge about mental health in pregnancy, unavailability of mental health services, and lack of a clear mental healthcare pathway [8].

## Limitations

The study findings should be evaluated in light of the following limitations. The cross-sectional design adopted in this study does not permit commentaries on causal relationship. The recruitment of participants with at least senior high school education for the study implies that the findings may not be applicable to those with less formal education. Therefore, the application of the study findings should be restricted to pregnant women who have attended at least secondary education in Ghana. As noted previously, most pregnant women start antenatal clinics in their second trimester, making it difficult to obtain a representation of this category of women. The findings could have been different with equal or similar number of pregnant women across the three trimesters. The use of convenience sampling increases the risk of selection bias that may impact the study result negatively. Indeed, because the participants were self-selecting, there is a possibility that only those with unique characteristics (e.g., interest in research) agreed to participate in the study. This line of reasoning further limits the application of the study findings to a broader population of pregnant women with at least senior secondary education. The choice of the study settings does not include government owned health facilities and it is not clear whether similar findings would be obtained from government facilities. Lastly, although single or few-item measures are capable of representing complex variables, multiple-item questionnaires are known to have superior psychometric properties, which could have influenced the findings reported here [38, 39].

## Conclusion

Pregnancy is a period of vulnerability for women to experience poor mental health. Several pregnancy-related stressors are unavoidable and many that are preventable continue to exist and manifest in diverse forms. Providing mental health support early enough to pregnant women could help to mitigate the impact of stressors on their mental and emotional well-being as well as the development of their children. While it is important to promote self-initiated mental health help-seeking, it is equally imperative to involve health professionals in the provision of mental health support, as advocated by the WHO's task shifting concept [40]. Both individual and health professional activation of the mental health pathway will ultimately benefit pregnant women and their unborn babies. With evidence suggesting high mental health illiteracy among pregnant women [41], greater responsibility is placed on health professionals to intervene in providing mental health services to pregnant women, as advocated by the NICE service management and clinical guidance for antenatal and postnatal mental health [16]. Health professionals should be well positioned to identify or detect pregnant women in need of mental health services and activate the pathway towards seeking help. The findings reported here re-emphasize the need for profiling pregnant women. To understand pregnant women who are likely to seek mental health support on their own, the study suggests that attention should be paid to those with the following background: History of hypertension or diabetes in the past and present pregnancies, victims of intimate partner violence, sleep difficulty, suicidal ideation and low social support. Health professionals can target pregnant women with the aforementioned experiences, in addition to those who feared vaginal delivery and expressed concern about COVID-19, to improve their mental and emotional well-being.

## Author Contributions

**Conceptualization:** Samuel Adjorlolo.

**Data curation:** Samuel Adjorlolo.

**Formal analysis:** Samuel Adjorlolo.

**Methodology:** Samuel Adjorlolo.

**Project administration:** Samuel Adjorlolo.

**Writing – original draft:** Samuel Adjorlolo.

**Writing – review & editing:** Samuel Adjorlolo.

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
