## [Decision Letter · Decision Letter 0]

8 Nov 2022

PONE-D-22-07154Seeking and Receiving Help for Mental Health Services Among Pregnant Women in GhanaPLOS ONE

Dear Dr. Adjorlolo,

Thank you for submitting your manuscript to PLOS ONE. After careful consideration, we feel that it has merit but does not fully meet PLOS ONE’s publication criteria as it currently stands. Therefore, we invite you to submit a revised version of the manuscript that addresses the points raised during the review process.

ACADEMIC EDITOR: Please insert comments here and delete this placeholder text when finished. Be sure to:Dear researchers. The reviewers have engaged with the paper and have suggested some reviews which are important in improving the paper.Please go review and respond to the comments as appropriate.==============================

We look forward to receiving your revised manuscript.

Kind regards,

Christmal Dela Christmals, PhD, MSc, BSc, RN

Academic Editor

PLOS ONE

Journal Requirements:

Reviewers' comments:

Reviewer's Responses to Questions

**Comments to the Author**

1. Is the manuscript technically sound, and do the data support the conclusions?

Reviewer #1: Yes

Reviewer #2: Yes

2. Has the statistical analysis been performed appropriately and rigorously? 

Reviewer #1: Yes

Reviewer #2: Yes

3. Have the authors made all data underlying the findings in their manuscript fully available?

Reviewer #1: Yes

Reviewer #2: Yes

4. Is the manuscript presented in an intelligible fashion and written in standard English?

Reviewer #1: Yes

Reviewer #2: Yes

5. Review Comments to the Author

Reviewer #1: The author has written on a very important subject titled “Seeking and Receiving Help for Mental Health Services Among Pregnant Women in Ghana”. This in no doubt adds to the body of knowledge in the subject area. It will however be important for the author to address the following concerns:

Abstract

Introduction

Line 2 under the introduction-periods of pregnancy should be prenatal and not perinatal as perinatal includes the postpartum period as well.

Methods

Ethical approval has been clearly stated.

Participants

What influenced the selection of the various proportions of the 702 pregnant women from the 4 different facilities indicated in the write-up.

What was the selection of the study facilities based on? The facilities selected were private, quasi government and CHAG hospitals. Do they not bias the patient population selected? Is this not one of the limitations of the study?

Study design and data collection

The inclusion criteria of nulliparity/multiparity means that parity was not a deciding factor in selection and therefore this can be deleted.

What sampling method was used to select the study participants?

Predictor variables

What is the validity and reliability of using these 2 questions 1 I have been belittled by my partner (i.e., abuse) and (2) I experienced no attention from my partner as items for measuring intimate partner violence.

Data analysis and strategy

Last paragraph on page 12, the sentence will be clearer if framed as “Results were entered as Crude Odds ratios (COR) and adjusted Odds ratios (AOR).”

Results

Participants

In the first paragraph of this section, the author states that participants were recruited with pregnancy throughout all the trimesters. This is also stated in the abstract under methodology.

It is not clear whether this selection was made with a plan at the beginning of the study and if this was the case, what was the exact plan to make sure that women were recruited from all the trimesters. Is it a case of what was realized after data was collected. This needs clarification.

Intable 2, under results, the author has a column captioned value. Please make this clear. Is it Chi square or p- square value?

With the legends under table 3, the author indicates + = point-biserial correlation coefficients. This legend cannot be found in the table.

In table 4, since the odds ratio and adjusted odds are both odds ratios, it is suggested that the bivariate odds is stated as Crude Odds ratio instead of just odds ratio.

The author states on page 15 under the predictors of mental health help seeking that “The statistical significance of the predictors did not change when demographic factors were controlled for.” However if you closely examine table 5, there are several instances when statistical significance changed after accounting for other factors in the regression model. For instance under past medical history as a predictor for offered mental health support, the crude odds ratio was 2.42*** (1.511- 3.870) which was significant but the adjusted odds ratio was 1.21(0.608-2.412) which was not significant. There are several of these in table 5.

Discussion

In the discussion on page 17 and 18 under Health Professionals Involvement in Promoting Mental Health in Pregnancy, the author indicates that “The documented relationship between low education and risk for poor mental health (Ten Kate et al., 2017) could partly explain why pregnant women with secondary school education in this study reported that health professionals enquired about their emotional and mental wellbeing.” Yet in the design of this study those with actual low level education, that is below secondary education level were left out due to language to English language as a barrier. The questionnaire could have been interpreted into other languages for them. The omission of that critical group is one of the key deficiencies of the study.

On page 19, the author indicates that:” Efforts to persuade pregnant women capable of vaginal delivery to do so may involve offering them mental health support, including counselling and providing referral sources to allay fear and anxiety”. Is the fear of vaginal delivery not better tackled as a comprehensive issue and not just a mental health issue? Issues to do with assurance of good monitoring during labour, issues to do with good pain relief etc.

Reviewer #2: The study was relevant to the current global economic challenges which resultant effects directly affect women and children. My comments were inserted and highlighted in the manuscript. These were minor corrections and some explanations that were required from you so that your article could be fit for this prestigious journal.

6. PLOS authors have the option to publish the peer review history of their article (what does this mean?). If published, this will include your full peer review and any attached files.

Reviewer #1: **Yes: **Kwaku Asah-Opoku

Reviewer #2: **Yes: **HAYFORD ISAAC BUDU

---

## [Author Response · Author response to Decision Letter 0]

13 Nov 2022

Seeking and Receiving Help for Mental Health Services Among Pregnant Women in Ghana

(PONE-D-22-07154)

Thank you for the reviewer comments. The response to the issues raised are highlighted in yellow ink for quick reference.

Reviewer #1’s Comments: 

Comment 1: The author has written on a very important subject titled “Seeking and Receiving Help for Mental Health Services Among Pregnant Women in Ghana”. This in no doubt adds to the body of knowledge in the subject area. It will however be important for the author to address the following concerns:

Response: Thank you very much for the positive feedback.

Abstract

Introduction

Comment 2: Line 2 under the introduction-periods of pregnancy should be prenatal and not perinatal as perinatal includes the postpartum period as well.

Response: This has been corrected. Thank you.

Participants

Comment 3: What influenced the selection of the various proportions of the 702 pregnant women from the 4 different facilities indicated in the write-up.

Response: Across the facilities, the decision was to recruit as many participants as possible. The potential influence of facility-level factors on the study variables was recognized, and statistical plan was to control for this effect. Thus, a minimum of 50 participants per facility was envisaged for any meaningful statistical comparison. 

Comment 4: What was the selection of the study facilities based on? The facilities selected were private, quasi government and CHAG hospitals. Do they not bias the patient population selected? Is this not one of the limitations of the study?

Response: Excellent observation. If you consider the study settings on the basis of ownership, it is apt to conclude on biasness. However, in terms of the demographics of participants who participated in the study, they are not entirely different from the general pool of pregnant women who attend government facilities. In fact, the health facilities serve pregnant from diverse socioeconomic background, an observation that mirror developments at government hospital. Nevertheless, we have included this as a potential limitation of the study as follows: The choice of the study settings does not include government owned health facilities and it is not clear whether similar findings would be obtained from government facilities.

Study design and data collection

Comment 5: The inclusion criteria of nulliparity/multiparity means that parity was not a deciding factor in selection and therefore this can be deleted. 

Response: This has been deleted.

Comment 6: What sampling method was used to select the study participants?

Response: Convenience sampling was used to recruit participants who met the inclusion criteria. The manuscript has been revised to accommodate this as follows; Recruitment of the participant was based on convenience, availability, and willingness to participate in the study.

Predictor variables

Comment 7: What is the validity and reliability of using these 2 questions 1 I have been belittled by my partner (i.e., abuse) and (2) I experienced no attention from my partner as items for measuring intimate partner violence.

Response: These items were extracted from existing tool used to screen for IPV. These two items produced an internal consistency using Cronbach formula of 0.52. This low internal consistency is expected given the small number of items, hence limiting the amount of variations needed to high or acceptable internal consistency.

Data analysis and strategy

Comment 8: Last paragraph on page 12, the sentence will be clearer if framed as “Results were entered as Crude Odds ratios (COR) and adjusted Odds ratios (AOR).”

Response: This have been incorporated into the revised manuscript.

Results

Participants

Comment 9: In the first paragraph of this section, the author states that participants were recruited with pregnancy throughout all the trimesters. This is also stated in the abstract under methodology. It is not clear whether this selection was made with a plan at the beginning of the study and if this was the case, what was the exact plan to make sure that women were recruited from all the trimesters. Is it a case of what was realized after data was collected. This needs clarification.

Response: The plan for data collection was to include pregnant women across the trimesters. The focus was to investigate how mental health problems in the prenatal period relate to pregnancy outcome, explaining why information on participant folder numbers were obtained for a follow-up. 

Comment 10: In table 2, under results, the author has a column captioned value. Please make this clear. Is it Chi square or p- square value?

Response: This has been corrected. The value represents chi-square value.

Comment 11: With the legends under table 3, the author indicates + = point-biserial correlation coefficients. This legend cannot be found in the table.

Response: In Table 3, + was attached to the continuous variables (e.g., social support, covid concern) to indicate that their correlations with the dichotomous variable was based on point-biserial correlations.

Comment 12: In table 4, since the odds ratio and adjusted odds are both odds ratios, it is suggested that the bivariate odds is stated as Crude Odds ratio instead of just odds ratio.

Response: This has been incorporated into the revised manuscript.

Comment 13: The author states on page 15 under the predictors of mental health help seeking that “The statistical significance of the predictors did not change when demographic factors were controlled for.” However if you closely examine table 5, there are several instances when statistical significance changed after accounting for other factors in the regression model. For instance under past medical history as a predictor for offered mental health support, the crude odds ratio was 2.42*** (1.511- 3.870) which was significant but the adjusted odds ratio was 1.21(0.608-2.412) which was not significant. There are several of these in table 5.

Response: Thank you for the excellent observation. This was an oversight. The statement has been deleted from the revised manuscript.

Discussion

Comment 14: In the discussion on page 17 and 18 under Health Professionals Involvement in Promoting Mental Health in Pregnancy, the author indicates that “The documented relationship between low education and risk for poor mental health (Ten Kate et al., 2017) could partly explain why pregnant women with secondary school education in this study reported that health professionals enquired about their emotional and mental wellbeing.” Yet in the design of this study those with actual low level education, that is below secondary education level were left out due to language to English language as a barrier. The questionnaire could have been interpreted into other languages for them. The omission of that critical group is one of the key deficiencies of the study.

Response: Thank you for the observation. You are right about the omission of persons with below secondary education. Unfortunately, resource issues prevented their inclusion. Second, the literature seems to have focused extensively on pregnant women with poor education. Those with relatively high education (the so-called elite pregnant women) have not been granted enough research attention with respect to mental health. The study is contributing uniquely to this context. This was acknowledged as a limitation of the study. 

Comment 15: On page 19, the author indicates that:” Efforts to persuade pregnant women capable of vaginal delivery to do so may involve offering them mental health support, including counselling and providing referral sources to allay fear and anxiety”. Is the fear of vaginal delivery not better tackled as a comprehensive issue and not just a mental health issue? Issues to do with assurance of good monitoring during labour, issues to do with good pain relief etc.

Response: Your observation is correct. A comprehensive approach is certainly needed. However, given the context of the study, we can speak to mental health support since other support systems or packages were not explored.

---

## [Decision Letter · Decision Letter 1]

3 Jan 2023

Seeking and Receiving Help for Mental Health Services Among Pregnant Women in Ghana

PONE-D-22-07154R1

Dear Dr. Adjorlolo,

We’re pleased to inform you that your manuscript has been judged scientifically suitable for publication and will be formally accepted for publication once it meets all outstanding technical requirements.

Kind regards,

Christmal Dela Christmals, PhD, MSc, BSc, RN

Academic Editor

PLOS ONE

Additional Editor Comments (optional):

Reviewers' comments:

Reviewer's Responses to Questions

**Comments to the Author**

1. If the authors have adequately addressed your comments raised in a previous round of review and you feel that this manuscript is now acceptable for publication, you may indicate that here to bypass the “Comments to the Author” section, enter your conflict of interest statement in the “Confidential to Editor” section, and submit your "Accept" recommendation.

Reviewer #1: All comments have been addressed

Reviewer #2: All comments have been addressed

2. Is the manuscript technically sound, and do the data support the conclusions?

Reviewer #1: Yes

Reviewer #2: Yes

3. Has the statistical analysis been performed appropriately and rigorously? 

Reviewer #1: Yes

Reviewer #2: Yes

4. Have the authors made all data underlying the findings in their manuscript fully available?

Reviewer #1: Yes

Reviewer #2: Yes

5. Is the manuscript presented in an intelligible fashion and written in standard English?

Reviewer #1: Yes

Reviewer #2: Yes

6. Review Comments to the Author

Reviewer #1: (No Response)

Reviewer #2: All comments have been addressed accordingly. The manuscript is fit for purpose;adding to existing knowledge,sharing information with the academic and the research public. Good work done by author(s).

7. PLOS authors have the option to publish the peer review history of their article (what does this mean?). If published, this will include your full peer review and any attached files.

Reviewer #1: No

Reviewer #2: **Yes: **HAYFORD ISAAC BUDU

---

## [Editor Report · Acceptance letter]

11 Jan 2023

PONE-D-22-07154R1 

Seeking and Receiving Help for Mental Health Services among Pregnant Women in Ghana 

Dear Dr. Adjorlolo:

I'm pleased to inform you that your manuscript has been deemed suitable for publication in PLOS ONE. Congratulations! Your manuscript is now with our production department. 

Kind regards, 

on behalf of

Dr. Christmal Dela Christmals 

Academic Editor

PLOS ONE